# Uniformity of Thermoluminescence and Optically Stimulated Luminescence Signals Over the Length of Doped LiMgPO_4_ Crystal Rods Grown by Micro-Pulling-Down Method

**DOI:** 10.3390/ma14010132

**Published:** 2020-12-30

**Authors:** Barbara Marczewska, Wojciech Gieszczyk, Mariusz Kłosowski, Marzanna Książek, Paweł Bilski, Łukasz Boroń

**Affiliations:** 1Institute of Nuclear Physics Polish Academy of Sciences, Radzikowskiego 152, 31-342 Krakow, Poland; Wojciech.Gieszczyk@ifj.edu.pl (W.G.); Mariusz.Klosowski@ifj.edu.pl (M.K.); Pawel.Bilski@ifj.edu.pl (P.B.); 2Department of Non-Ferrous Metals, AGH University of Science and Technology, 30 Mickiewicza Ave., 30-059 Krakow, Poland; mksiazek@agh.edu.pl; 3Łukasiewicz Research Network, Łukasiewicz—Krakow Institute of Technology, 73 Zakopianska St., 30-418 Krakow, Poland; Lukasz.boron@kit.lukasiewicz.gov.pl

**Keywords:** micro-pulling-down method, LiMgPO_4_ crystals, Tm or Tb doping, TL, OSL

## Abstract

The purpose of this work was to systematically assess the distribution of thermoluminescence (TL) and optically stimulated luminescence (OSL) properties along the length of crystals grown by the micro-pulling-down method, in relation to the microstructure and distribution of activators. We analyzed lithium magnesium phosphate (LiMgPO_4_; LMP) crystals doped with terbium (Tb; 0.8 mol%) or with a combination of thulium (Tm; 0.8 mol% or 1.2 mol%) and boron (B; 10 mol%). Crystals of several centimeters in length and 3 mm in diameter were cut into 20–40 slices, depending on length. For each sample, TL glow curves and OSL decay curves were examined. Optical microscopy and scanning electron microscopy were used to assess the microstructure and elemental composition of several selected samples. Two-dimensional TL readouts were performed to identify the areas with higher and lower signal emission. Our study showed that there may be significant differences not only in LMP sensitivity along the crystal’s axis but also on the surface of the slice. The distribution of activators varies depending on the type of crystals and strongly affects luminescence properties.

## 1. Introduction

The micro-pulling-down (MPD) technique is a relatively new method of crystal growth that was developed in the 1990s in Fukuda Laboratory, Sendai, Japan. In contrast to the well-known Czochralski method, in which the crystal is pulled up after the seed touch, the MPD method involves pulling of the melted material in a downward direction through a microcapillary channel made in the bottom of a crucible [1,2]. This allows crystals to be grown at a relatively high speed and to obtain them within a few hours. The method also provides an excellent opportunity to quickly assess the effects of a change in doping on the properties of the modified material.

Crystals obtained by MPD methods are in the form of rods with a diameter of several millimeters and a length of several centimeters. The rods are commonly cut into smaller pieces with a wire saw. Usually, the feed raw material is not chemically homogeneous; rather, it constitutes a chemical compound or a mixture of compounds. Even if it is a single compound, it might be doped with other elements. The uniform distribution of admixtures in the input raw material does not remain constant during melting or during the crystal extraction process itself. It is hypothesized that there may be differences in the luminescence properties of individual slices cut out from the various sites of the same crystal rod obtained by the MPD method. Research is currently ongoing to elucidate this effect.

We examined the uniformity of luminescence properties along crystals grown by the MPD method, using a sample of lithium magnesium phosphate (LiMgPO_4_; LMP), which is one of the several relatively new luminescent materials that could be applied in dosimetry due to high sensitivity to ionizing radiation. The LMP compound was shown to have a high radiosensitivity and a broad linear dose–response range [3]. Numerous research articles on luminescent properties of LMP doped with different elements have been published since 2010 [3,4,5,6,7,8,9,10,11,12,13,14,15,16,17,18,19,20,21,22,23,24,25,26,27,28,29,30,31]. The LMP crystal codoped with terbium (Tb) and boron (B) has been most extensively studied, but other rare earth dopants such as Sm, Tm, and Eu have also been tested [5,10,13,20].

Usually, individual samples are tested separately because it is assumed that the properties are the same for the whole crystal. However, the problem arises when a group of samples has to be examined to determine, for example, a dose response in a dose range or a signal change over time after irradiation by taking individual samples from the sample batch for each measuring point. The characteristics of individual samples may vary and may affect the measured values. This was also important when radioluminescence was tested using a single crystal rod to obtain samples for measurement [29]. To our knowledge, no such studies on LMP crystals have been conducted so far. Therefore, it seems crucial to investigate the properties of individual parts of crystal rods.

The aim of the present work was to systematically examine the individual parts of crystals grown by the MPD method with regard to thermoluminescence (TL) and optically stimulated luminescence (OSL) properties along the length of the crystals. The study was conducted using pieces of LMP crystals doped with Tb (0.8 mol%) and a combination of Tm (0.8 mol% or 1.2 mol%) and B (10 mol%). Thermoluminescence properties were investigated both with a conventional TL reader and a two-dimensional (2D) reader equipped with a charge-coupled device (CCD) camera. The structural properties as well as chemical and phase composition of LMP were examined by light microscopy and scanning electron microscopy.

## 2. Materials and Methods

Crystals were grown by the MPD method using a Cyberstar facility. The preparation of raw material and the process of crystal growth were described in detail previously [26]. The starting feedstock for crystal production was LMP powder prepared according to the standard procedure of a solid-state reaction in air. Lithium hydroxide (LiOH), hexahydrate magnesium nitrate (Mg(NO_3_)_2_·H_2_O), and ammonium dihydrogen phosphate (NH_4_H_2_PO_4_), were used as substrates. The solid-state chemical reaction was interrupted by several annealing cycles at a temperature ranging from 200 °C to 750 °C and to mix the reaction products. Boric acid (H_3_BO_3_), or borax (Na_2_B_4_O_7_·10H_2_O),was used for doping the phosphors with B ions, while Tb_4_O_7_, and Tm_2_O_3_, oxides were used for doping the phosphors with Tb and Tm ions, respectively. The chemical composition and concentrations of dopants are given in Table 1.

The raw materials were loaded into the graphite crucible and melted inside an inductive furnace. The molybdenum overlay was placed around the crucible to improve the heating conditions and thermal energy transfer to the raw material. A graphite after-heater, and two layers of alumina ceramic thermal isolation, were also applied to ensure an appropriate temperature gradient within the growth zone. Finally, the melt was pulled down, using an iridium seed, at a constant rate of 0.2 mm/min in inert gas atmosphere (Ar). The obtained rod-shaped crystals had a diameter of approximately 3 mm and a length of up to 60 mm.

In a previous experiment [26], an X-ray diffraction analysis was systematically conducted in a group of 14 LMP crystals: without admixture and with addition of Tb, Tm, and B in different proportions. The samples were powdered before the analysis. The study showed trace amounts of TbPO_4_, TmPO_4_, and MgO phases in Tb-doped LMP (0.8 mol%) and Tm-doped LMP (0.8 mol%), but no phases other than LiMgPO_4_ were observed in samples doped with B (either 1% or 10%).

The crystals were cut using a diamond wire, saw into 1-mm–thick slices, numbered consecutively starting from the beginning of the crystal (from the seed side). An exemplary crystal with marked sample numbering is presented in Figure 1.

Next, the samples were examined for their TL and OSL properties. Standard TL and OSL measurements were performed in an automatic TL/OSL-DA20 reader (Risø, Roskilde Denmark) with blue LED stimulation (470 nm), a photomultiplier tube (EMI 9235QB), and an Sr-90/Y-90 beta source used for irradiation. Detailed specifications of the reader were recently described by Bilski et al. [30] and Wróbel et al. [31]. For OSL measurements, a Hoya U-340 filter was used, whereas Hoya U-340 and Schott BG39 filters were applied for TL measurements. The LMP samples were annealed by heating them up to 500 °C at a constant heating rate of 5 °C/s. This was followed by bleaching by the OSL measurement with blue LED stimulation (90% power) for 600 s in the Risø reader before use. Standard TL readouts of the irradiated LMP crystals were carried out at a constant heating rate of 2 °C/s from room temperature up to 500 °C, while standard OSL measurements were performed at room temperature, with 90% power blue LED stimulation for 600 s. The TL and OSL signals were normalized to the weight of the crystals. The samples were bleached in the Risø reader between each measurement, using the following procedure: initially, the OSL readout for 100 s, then twice the TL readout up to 500 °C (heating rate of 5 °C/s), and then the OSL readout again for 100 s. All the measured TL glow curves and OSL decay curves were then processed using dedicated GlowVIEW software developed by Gieszczyk and Bilski in 2017 [32].

Two-dimensional TL readouts were performed with a reader with a CCD camera, which was constructed at the Institute of Nuclear Physics Polish Academy of Sciences [33,34,35,36]. The TL reader is equipped with a 60-mm diameter steel heater, the temperature of which can be raised linearly up to 400 °C, with heating rates ranging from 1 °C/s to 10 °C/s. The signal is recorded by a PCO SensiCam TM VGA CCD 12-bit camera, which registers the light signal with a dynamic range of up to 4500 counts per pixel. The samples were read out at a heating rate of 3 °C/s. The measurements consisted of recording the light emitted from the samples integrated over the entire temperature range up to 350 °C. After being heated up to 350 °C, the samples were slowly cooled down to room temperature. Each sample was irradiated with 1-kGy beta-particle dose to obtain TL glow curves (TL intensity vs. temperature) from selected areas of the sample by recording the sequential images every 2 s.

The microstructure observations were conducted using an AxioObserver Zm1 tabletop metallographic microscope (Carl Zeiss), for observation in a bright field, dark field, polarized light, and interference contrast. The images were recorded with the AxioVisio software.

The chemical composition of doped LMP crystal slices was analyzed by a scanning electron microscope SCIOS FEI, equipped with an energy-dispersive spectrometer using polished specimen cross-sections normal to the surface of the slices. The samples were included in resin and sputtered with gold.

## 3. Results

### 3.1. TL Signal along Crystal Length

The first step was to cut the slices perpendicular to the crystal axis and assign them numbers starting from the initial segment of the newly grown crystal, yielding 27, 25, and 18 samples for the LMP:Tb, LMP:Tb (0.8 mol%), B (10 mol%) and LMP:Tb (1.2 mol%), B (10 mol%) crystal, respectively (see Table 1). Next, the samples were placed on the reader carousel of the Risø reader, and TL and OSL measurements were performed. After taking the TL reading and before subsequent irradiation, the samples were bleached, first by the OSL readout with blue diodes for 100 s, then twice annealed by the TL readout up to 500 °C with the heating rate of 5 °C/s and then again by the OSL readout with blue diodes for 100 s.

The glow curves measured for the Tb-doped samples are shown in Figure 2A. It is visible that these curves differ significantly from each other and can be divided into two groups regarding to their shape. The first group contains the curves measured for the slices from 1 (beginning of the crystal) to 12, and the second group contains the curves measured for the samples from 13 to the end of the crystal. The characteristic feature of the glow curves measured for the first group is the presence of three, well-separated thermally, peaks with maxima at around 100–125 °C, 180–220 °C, and 325–400 °C. The samples included in the second group exhibit more similar glow curve shape with similar peaks of relatively low amplitudes peaked at around 125 °C and 280–325 °C. The different shapes of the measured glow curves may suggest that different mechanisms of impurities incorporation were involved in the growth of different parts of the crystal. These are also correlated with the changes of the energetic distribution of electron traps in consecutive slices of crystal what is strongly connected to the shape of the measured glow curves.

In contrast to the curves described above, all curves for the samples of Tm-doped LMP crystals had a similar shape regardless of the Tm concentration (0.8 mol% or 1.2 mol%) and the part of the crystal where the samples were obtained from. The only difference was observed for the main peak amplitude (Figure 3A). In this case, the most prominent peak was in the temperature range between 300 °C and 380 °C. The peaks of TL glow curves registered for higher Tm doping (1.2 mol%) had a significantly lower amplitude.

Figure 2B and Figure 3B show the TL signal integrated over the temperature range from room temperature to 400 °C, calculated for the consecutive slices cut from corresponding crystals (Tb-doped in Figure 2B and Tm,B-doped in Figure 3B). All samples were weighed before the measurements and each value has been normalized to the sample’s weight. As can be seen in these Figures, the integrated TL signal differs significantly from sample to sample. Interestingly, the most unambiguous results were obtained for the Tb-doped sample. Namely, the initial part of the crystal showed the highest luminescence intensity, up to 4–5 times higher as compared to the signal of samples originating from the end of the crystal (Figure 2B). Moreover, the distribution of values evaluated for the slices in-cluded in the first group (slices from 1 to 12) is very heterogeneous (widely spread) in comparison to the distribution of values evaluated for the samples included in the second group (slices from 13 to the end of the crystal), which is quite uniform. In the case of Tm,B-doped crystal (Figure 3B), significant differences in the TL signal intensities are also visible, however, these cannot be directly correlated to the part (beginning or end) of the crystal—it cannot be surely said that the initial part of the crystal shows higher lumines-cence intensity than the end of the crystal due to the very large spread of points (increasing trend). Also, there is no such great difference between the beginning and end part of the crystal. This may be related to the differences in Tb and Tm incorporation mechanism in the LMP host matrix.

### 3.2. OSL Signal along Crystal Length

Similarly, OSL decay curves were recorded in the Risø reader after irradiation of all samples with 0.2 Gy of beta rays. The OSL decay curves for LMP doped with Tb (0.8 mol%) can be classified into two groups: those with a higher signal (curves 1–10) and those with a lower signal (curves 11–25). The typical decay curves for the first group (curve for slice 2) and for the second group (curve for slice 19) are presented in Figure 4A. The total OSL signals integrated from 0 to 300 s for all samples are shown in Figure 4B. The integrated OSL signals showed a similar trend as the TL signal (Figure 2B), which means that the highest TL and OSL signals were found for the samples with the same numbers. Two characteristic decay curves (for slices number 7 and 22) are shown in Figure 5A, while integrated OSL signals (from 0 to 300 s) for all LMP samples doped with 0.8 mol% Tm and 10 mol% B are presented in Figure 5B. In this case, the sum of TL and OSL signals for individual samples along the crystal length was also similar. The total OSL signals for samples doped with 1.2 mol% Tm and 10 mol% B are presented in Figure 6. Systematic measurements of the luminescence signal revealed significant differences between the individual parts of the crystal along its length (even four to five times), irrespective of the method (TL or OSL).

### 3.3. TL Signal on the Surface of Crystal Slices (2D TL)

In the case of conventional TL measurements, the signal is collected from the whole surface of the sample and is treated as a sum of the total TL signal emitted by the sample. By replacing the photomultiplier tube with a CCD camera, an image of the TL signal distribution on a given surface can be obtained. Two-dimensional TL readouts were performed in a special TL reader constructed specifically for the measurement of the dose distribution on homogeneous samples. However, after irradiation with an equal dose of the entire surface, it also allows an observation of the sites both more and less sensitive to radiation [29,30,31,32].

Measurements in the 2D TL reader were taken from the samples with the highest and lowest TL and OSL signals. The readouts were performed with a heating rate of 5 °C/s, from room temperature up to 350 °C, after irradiating the samples with 20 Gy in a homogeneous field of radiation. As shown in Figure 7, Figure 8 and Figure 9 there were areas with a higher or lower emission of the TL light on the surface of the samples. Only sample #12 was more homogeneous (Figure 7b). The TL reading control program allowed us to read the curves from any selected area of the sample. The TL glow curves for the chosen areas of the samples are presented in Figure 10 and Figure 11. In the case of a Tb-doped sample, the curves were different for the two slices of the crystal. The first one (sample No. 3 with a higher total TL signal) had two peaks at a temperature of 100 °C and 135 °C and a shoulder with a curve inflection at 175 °C and 225 °C. On the other hand, the TL curves of sample #16 had three peaks at 75 °C, 180 °C, and 280 °C (all of low amplitude compared with sample No. 3). The changes in the shape of the curves indicate a different mechanism of admixture embedding in the structure and a different electron trap mechanism in different parts of the crystal.

In contrast, the samples doped with Tm and B, independent of the Tm concentration, presented one evident peak around the temperature of 225 °C, the amplitude and peak position of which changed depending on the sample and location on the sample (Figure 8 and Figure 9). The shape of this TL glow curve was similar to that obtained in a conventional TL reader (Figure 3A).

### 3.4. Light Microscopy

Evident differences in the TL signals of individual surface areas led us to examine the microstructure of the samples. The samples were embedded in resin, polished, and subjected to light microscopic examination, using a range of magnification from 50 to 1000 times. The analysis of microstructures revealed more solidified ceramics than that observed for single crystals. Instead of the expected single-crystal structure, we observed a more complex multiphase substructure in all the samples, irrespective of whether they were doped with Tb or Tm (Figure 12, Figure 13 and Figure 14). Between “smooth” areas, which may indicate the presence of one phase of LiMgPO_4_, there are “hieroglyphic” areas that resemble eutectic alloys [37]. It is possible that when the crystal pulling process is too fast, thermal conditions result in crystal solidification, which produces multiphase eutectic-like inclusions. A comparison of the microstructure between samples with higher and lower sensitivity to ionizing radiation for a given type of crystal did not reveal any significant differences. Such differences might result from the local distribution of multiphase inclusions that affect sensitivity. The multiplicity of phases in these areas indicates a highly complex system of phase coexistence; therefore, a study of the elemental composition of individual areas was necessary.

### 3.5. Scanning Electron Microscopy

To determine the chemical elemental composition, especially Tb and Tm distribution in the microregions of the sample surface, energy-dispersive spectroscopy (EDS) was used. The picture and the corresponding region with a higher content of Tb (brighter points) is shown in Figure 15. The sample contains areas with higher and significantly lower amounts of Tb.

In contrast to LMPs doped with Tb, those doped with Tm (irrespective of the concentration) showed Tm presence in the whole investigated microregion, although there were visible areas with a larger amount of Tm (Figure 16 and Figure 17).

## 4. Conclusions

Lithium magnesium phosphate crystals doped with Tb and Tm are considered as phosphors that are highly sensitive to ionizing radiation and as such are candidates for potential dosimetry material in the future. Our systematic TL and OSL investigation of Tb- and Tm-doped LMP crystal rods grown by the MPD method confirmed their high sensitivity. However, it also revealed significant differences in the luminescent properties between particular samples obtained from different parts of the crystals. Differences in the sensitivity of the individual areas of the crystal can change even up to eight times. In the Tm-doped crystal, a rather random spread of sensitivity over the length of the crystal can be observed, but in the case of Tb doping, the part of the crystal obtained at the beginning of the growth process showed higher sensitivity than the remaining parts. Conventional and 2D TL readouts of Tb-doped samples revealed areas with different sensitivity to ionizing radiation on the surface of the sample cut out perpendicular to the crystal axis. These areas are characterized also by a different shape of TL curves. This indicates a different mechanism of admixture embedding in the structure and a different electron trap mechanism in various parts of the crystal. The microscopic examination showed a highly complex multiphase microstructure of LMPs, which looks more like solidified ceramics than a crystal structure. In order to obtain crystals with a more homogeneous structure, certain adjustments might be necessary. Moreover, crystals could be grown at different speeds, and special heat treatment or multiple remelting operations could be applied. Therefore, further research in this area is required.

## Figures and Tables

**Figure 1 materials-14-00132-f001:**
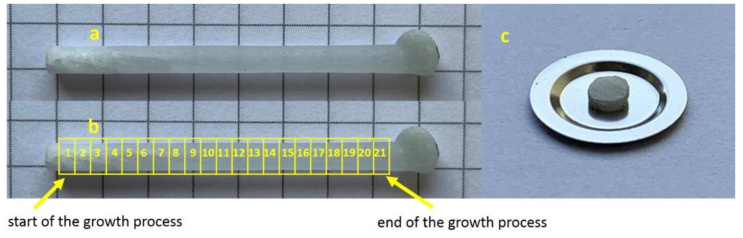
LMP crystal doped with Tb (0.8 mol%) (**a**), sites of sampling along the crystal axis (**b**) and a slice of the crystal placed on a Risø cup (**c**).

**Figure 2 materials-14-00132-f002:**
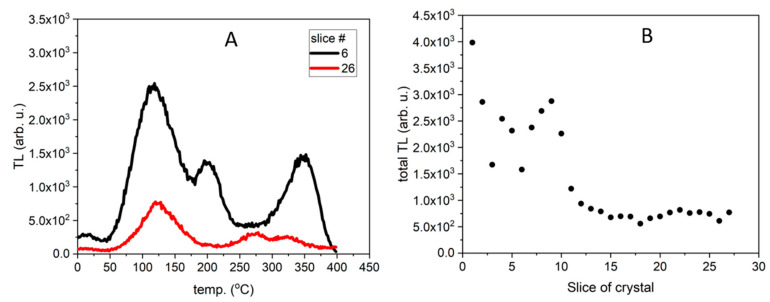
TL signal of LMP doped with 0.8-mol% Tb: (**A**)—TL glow curve for slice #6 is typical for the curves for slices 1–12, curve for slice #26 is typical for the curves for slices 13–27. (**B**)—Integrated TL signal in the temperature range between 20 °C and 400 °C calculated for all TL curves. The samples were irradiated with 0.2 Gy of beta rays in the Risø reader. Slice of crystal—numbers represent labelling of samples along the crystal length, as illustrated in Figure 1.

**Figure 3 materials-14-00132-f003:**
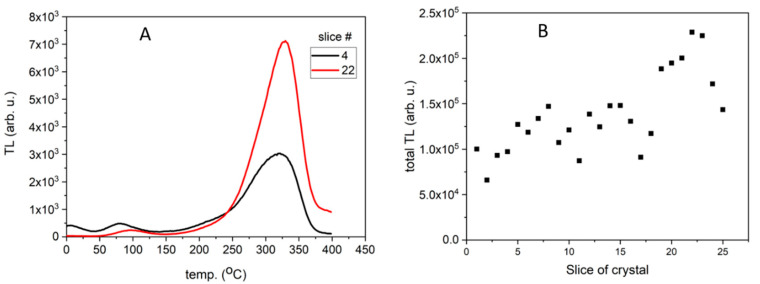
TL signal of LMP doped with 0.8-mol% Tm and 10-mol% B. (**A**)—TL glow curves for slices #4 and #22. (**B**)—Integrated TL signal in the temperature range between 20 °C and 400 °C calculated for all TL curves. The samples were irradiated with 0.2 Gy of beta rays in the Risø reader. Slice of crystal—numbers represent labelling of samples along the crystal length, as illustrated in Figure 1.

**Figure 4 materials-14-00132-f004:**
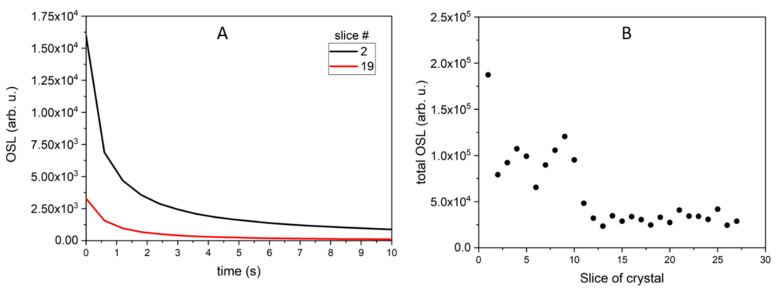
OSL signal of LMP doped with 0.8-mol% Tb. (**A**)—OSL decay curves for slices #2 and #19, typical for all slices, differing only in amplitude. (**B**)—total OSL signals integrated from 0 to 300 s for all OSL curves. The samples were irradiated with 0.2 Gy of beta rays in the Risø reader. Slice of crystal—the numbers represent labelling of samples along the crystal length, as illustrated in Figure 1.

**Figure 5 materials-14-00132-f005:**
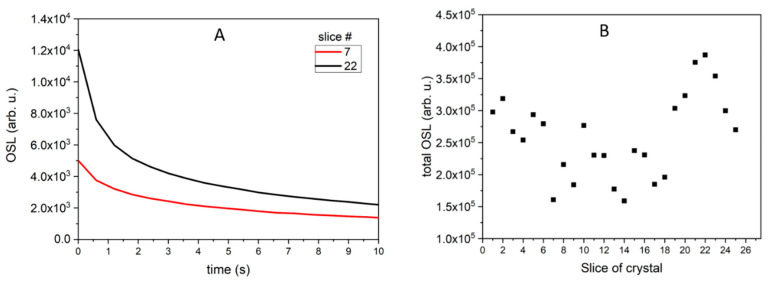
OSL signal of LMP doped with 0.8-mol% Tm and 10-mol% B. (**A**)—OSL decay curves for slices #7 and #22. (**B**)—total OSL signals integrated from 0 to 300 s for all OSL curves. The samples were irradiated with 0.2 Gy of beta rays in a Risø reader. Slice of crystal—the numbers represent labelling of samples along the crystal length, as illustrated in Figure 1.

**Figure 6 materials-14-00132-f006:**
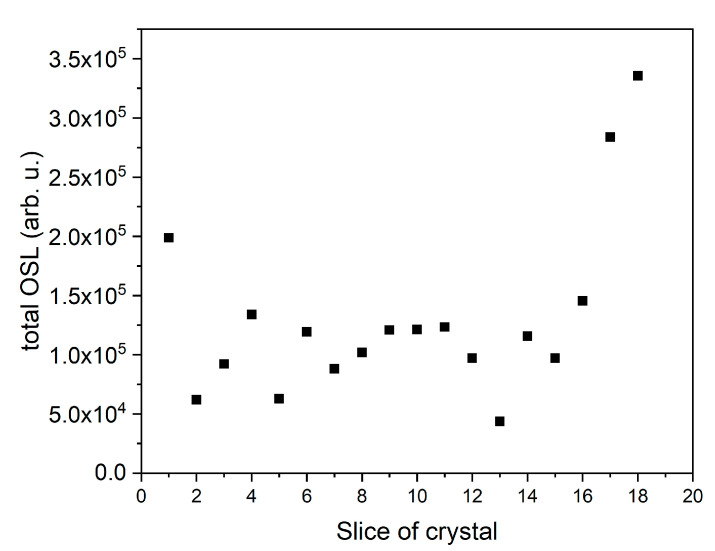
OSL signal of LMP doped with 1.2-mol% Tm and 10-mol% B integrated from 0 to 300 s for all OSL curves. The samples were irradiated with 0.2 Gy of beta rays in a Risø reader. Slice of crystal—the numbers represent labelling of samples along the crystal length, as illustrated in Figure 1.

**Figure 7 materials-14-00132-f007:**
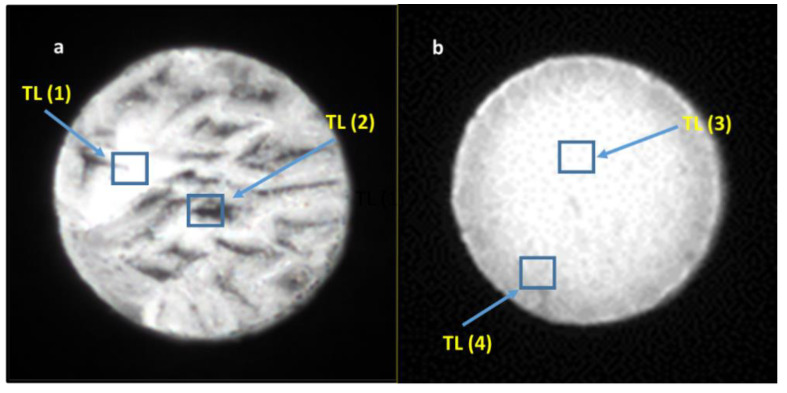
Images of luminous surfaces of samples No. 3 (**a**) and No. 12 (**b**) cut out from the LMP crystal doped with 0.8-mol% Tb. The images were recorded in a reader with a CCD camera after samples were irradiated with 20 Gy of beta rays.

**Figure 8 materials-14-00132-f008:**
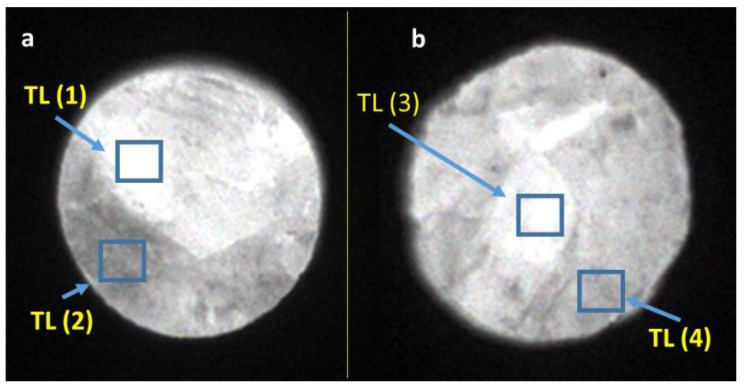
Images of the luminous surface of samples No. 14 (**a**) and No. 21 (**b**) cut out from the LMP crystal doped with 0.8-mol% Tm and 10-mol% B. The images were recorded in a reader with a CCD camera after irradiation with 20 Gy of beta rays. Marked boxes correspond to the sites of TL signal measurement.

**Figure 9 materials-14-00132-f009:**
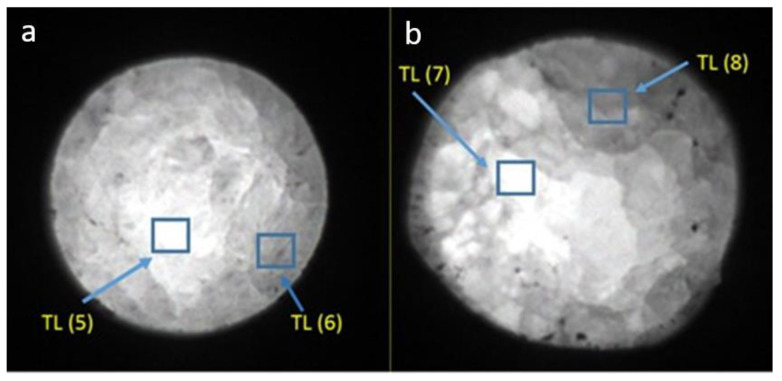
Images of the luminous surface of samples No. 7 (**a**) and No. 17 (**b**) cut out from the LMP crystal doped with 1.2-mol% Tm and 10-mol% B. The images were recorded in a reader with a CCD camera after irradiation with 20 Gy of beta rays. Marked boxes correspond to the sites of TL signal measurement.

**Figure 10 materials-14-00132-f010:**
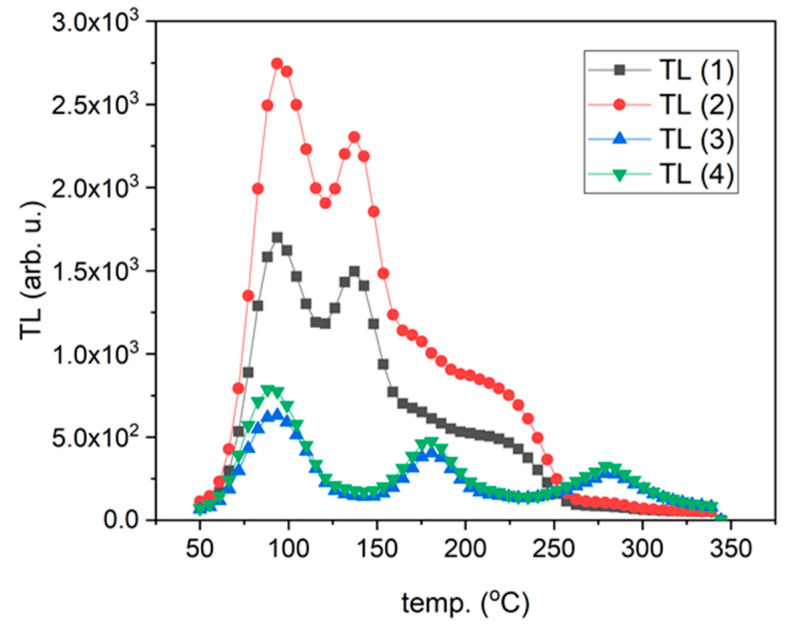
TL glow curves of LMP doped with 0.8-mol% Tb recorded for areas selected from the surface of the samples (indicated in Figure 7).

**Figure 11 materials-14-00132-f011:**
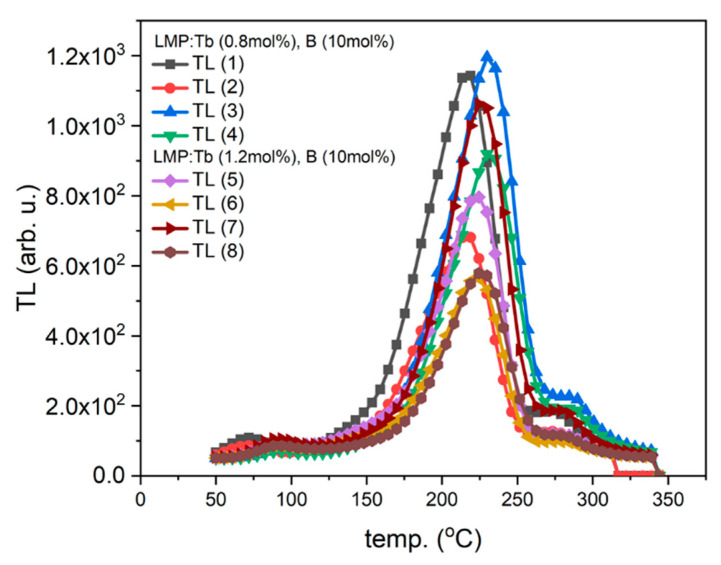
TL glow curves of LMPs doped with Tb and B. TL (1)–TL (8) curves were recorded for areas selected from the surface of the samples (marked boxes indicated in Figure 8 and Figure 9).

**Figure 12 materials-14-00132-f012:**
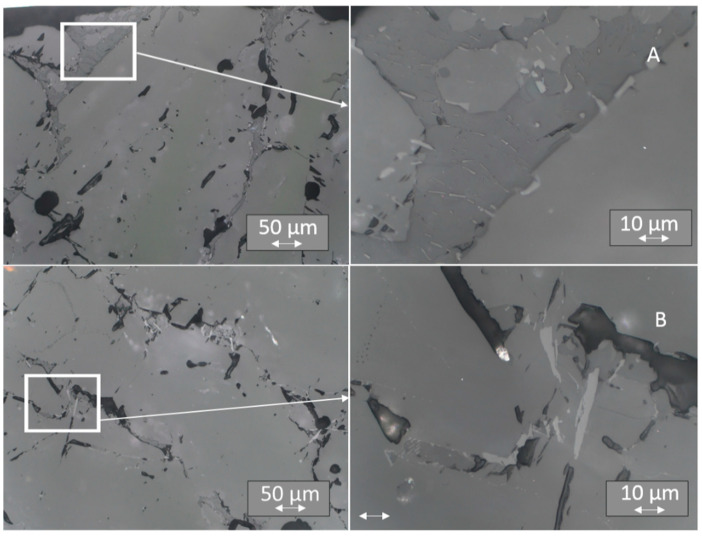
Microstructure of LMP doped with 0.8-mol% Tb. Top panels (**A**)—sample No. 2 at a magnification of 200× (on the left) and 1000×(on the right); bottom panels (**B**)—sample No. 11 at a magnification as above. The inset on the left has been further magnified on the right.

**Figure 13 materials-14-00132-f013:**
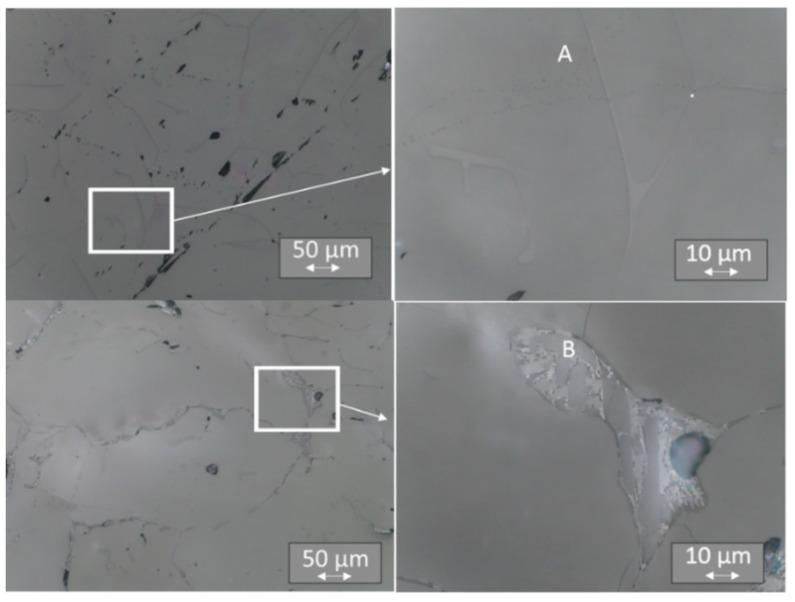
Microstructure of LMP doped with 0.8-mol% Tm and 10-mol% B. Top panels (**A**)—sample No. 7 at a magnification of 200× (on the left) and 1000× (on the right); bottom panels (**B**)—sample No. 22 at a magnification as above. The inset on the left has been further magnified on the right.

**Figure 14 materials-14-00132-f014:**
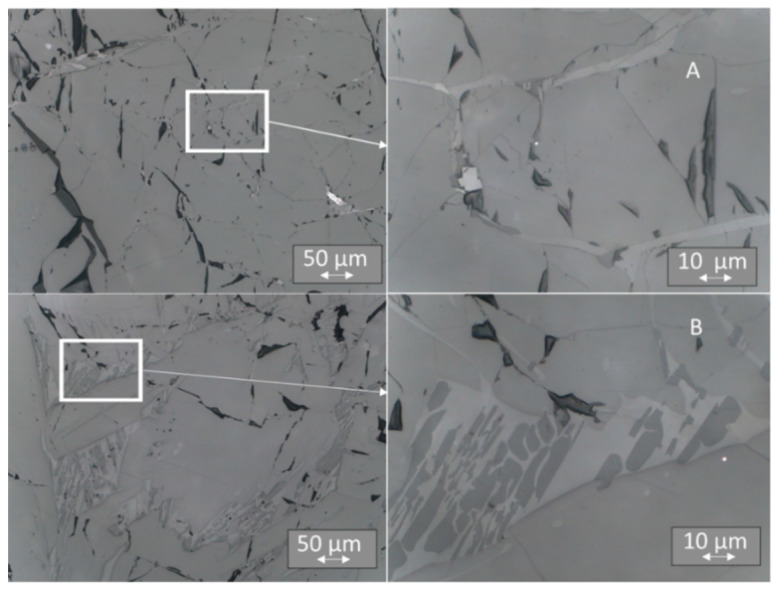
Microstructure of LMP doped with 1.2-mol% Tm and 10-mol% B. Top panels (**A**)—sample No. 5 at a magnification of 200× (on the left) and 1000× (on the right); bottom panels (**B**)—sample No. 18 at a magnification as above. The inset on the left has been further magnified on the right.

**Figure 15 materials-14-00132-f015:**
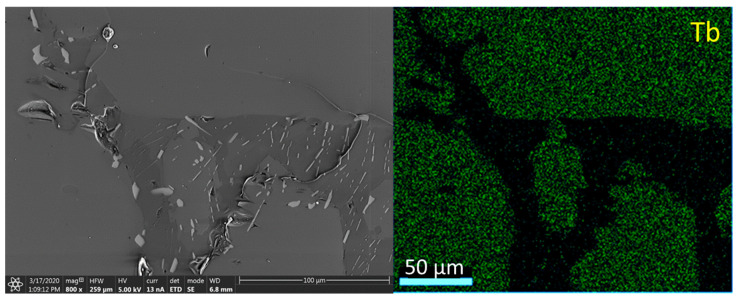
Sample presentation of the energy-dispersive spectroscopy analysis of the area from sample No. 2 of LMP doped with 0.8-mol% Tb, with mapping of the Tb distribution (right-hand panel).

**Figure 16 materials-14-00132-f016:**
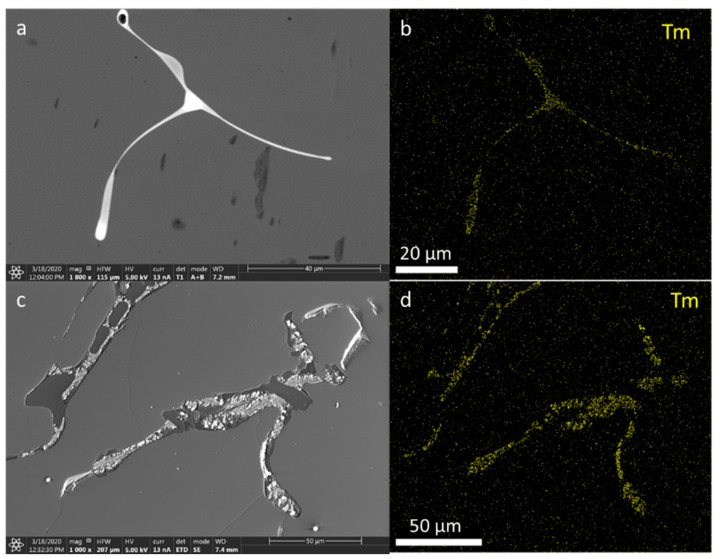
Sample presentation of the energy-dispersive spectroscopy analysis of the area from samples of LMP doped with 0.8-mol% Tb and 10-mol% B, with mapping of the Tm distribution (right-hand panels); top panels—sample No. 7, bottom panels—sample No. 22.

**Figure 17 materials-14-00132-f017:**
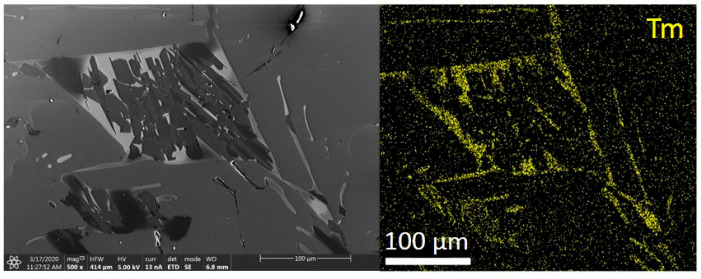
Sample presentation of the energy-dispersive spectroscopy analysis of the area from sample No. 18 of LMP doped with 1.2-mol% Tb and 10-mol% B, with mapping of the Tm distribution (right-hand panel).

**Table 1 materials-14-00132-t001:** Chemical composition and concentration of dopants of lithium magnesium phosphate (LMP) crystals and slice numbers tested by a given method.

Type and Concentration of Dopants in LiMgPO_4_ (mol%)			Slice Numbers Tested by A Given Method		
Tb	B	Tm	TL, OSL	2D TL	LM, SEM
0.8	-	-	all (27 slices)	3, 16	2, 11
-	10	0.8	all (25 slices)	14, 21	7, 22
-	10	1.2	all (18 slices)	7, 17	5, 18

## Data Availability

Data sharing is not applicable to this article.

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
