# Peer review of "Uniformity of Thermoluminescence and Optically Stimulated Luminescence Signals Over the Length of Doped LiMgPO4 Crystal Rods Grown by Micro-Pulling-Down Method"

_materials, 2020, doi:10.3390/ma14010132_

Round 1

Reviewer 1 Report

It was a pleasure for me to review the manuscript entitled “Uniformity of TL and OSL signals over the length of the doped LMP crystal rods grown by micro-pulling down method".  In particular, optically stimulated luminescence properties were investigated, including thermoluminescence for two type of LiMgPO4 crystal.

In my opinion, the present version is not adequate for the publication in Materials journal. If I may, an improvement is need before the publication. For example, there are many inaccuracies. Moreover, often the manuscript is too qualitative, and sometimes is not enoght clear.

In any case, I have some comments which should be addressed before the publication. I hope they will be helpful to further improve the manuscript.

Title: TL, OSL, LMP are not defined yet: I suggest to write, theromoluminescence,  optically stimulated luminescence,  LiMgPO4 explicitly.

Line 47-50: Is not reasonable quote, as a result, something not yet published. Maybe, can be write something as: The luminescence properties of […] could be different; investigation about this effect is in progress.

Fig. 1: It is interest show also one picture for each crystal studied in the manuscript.

Line 135: the title could be improved, e.g.: “TL and OSL measurements” (?) or maybe “TL and OSL spectra of  LiMgPO4(Tb),(Tm,B)”

Line 136: it is useful to indicate the total number of slices per each crystal.

General comment for the figures: often the image-quality is too poor and the information in the caption and/or in the text is not sufficient.

Fig 2A: specify what is the meaning of numbers reported in the legend. Why some numbers are missed?  

Fig. 2C: very bad quality. Specify in the caption if the curves are, for example, different broken-lines that connect the experimental points.

The paragraph 142 – 151 is not clear: it is strongly necessary to improve it.

Line 155 “[…] the main and only peak […]”: Fig. 3A shows two peaks, and in any case could be possible also a third component between the two external ones. It is necessary to reconsider the comment and the interpretation.

The paragraph 158 – 167 is not clear: it is strongly necessary to improve it.

General comment about the 3.1 section.

Maybe, It could be useful to separate the 3.1 section (TL and OSL) in two sections: a section dedicates to the TL measurements, and the second one for the OSL measurements. However, the interpretation of the results should be improved.

Figs 5 and 8: indicate, in the captions, what is the meaning of TL(1), .., TL(4),…,TL(8)  in the caption and their link with Figs 6 and 9 respectively.

Line 227: “only one high-temperature peak” --> “one evident peak around the temperature of 225 °C”

Line 227-228: also the peak position changes slightly depending on the samples and the zone of the sample. It is better to add this information in the text.

In all the manuscript, it is very hard to follow what is the crystal-slice used for the different measure reported in the text or in the figures. It is necessary to find a strategy to improve the reading.

The style of references seem not in the MDPI format: please check it.

Author Response

Dear Editor and Reviewers,

The authors are grateful to the reviewers for all comments. The responses to reviewers' comments are provided below. All changes to the text are marked in red. The article has been linguistically checked.

Reviewer #1

It was a pleasure for me to review the manuscript entitled “Uniformity of TL and OSL signals over the length of the doped LMP crystal rods grown by micro-pulling down method".  In particular, optically stimulated luminescence properties were investigated, including thermoluminescence for two type of LiMgPO4 crystal.

In my opinion, the present version is not adequate for the publication in Materials journal. If I may, an improvement is need before the publication. For example, there are many inaccuracies. Moreover, often the manuscript is too qualitative, and sometimes is not enoght clear.

In any case, I have some comments which should be addressed before the publication. I hope they will be helpful to further improve the manuscript.

Detailed remarks:

Title: TL, OSL, LMP are not defined yet: I suggest to write, theromoluminescence,  optically stimulated luminescence,  LiMgPO4 explicitly.

It has been corrected.

Line 47-50: Is not reasonable quote, as a result, something not yet published. Maybe, can be write something as: The luminescence properties of […] could be different; investigation about this effect is in progress.

It has been corrected in the text.

Fig. 1: It is interest show also one picture for each crystal studied in the manuscript.

We wanted to include photos of individual samples but the samples look very similar and that's why we decided to put the photo only of one sample.

Line 135: the title could be improved, e.g.: “TL and OSL measurements” (?) or maybe “TL and OSL spectra of  LiMgPO4(Tb),(Tm,B)”

According to your suggestions, we have divided this chapter into two parts „3.1 TL signal along crystal length” and „3.2 OSL signal along crystal length”. The next chapter has the title „3.3 TL signal on the surface of the crystal slices (2-D TL)”

Line 136: it is useful to indicate the total number of slices per each crystal.

We added: „ yielding 27, 25, and 18 samples for the LMP:Tb, LMP:Tb (0.8 mol%), B (10 mol%) and LMP:Tb (1.2 mol%), B (10 mol%) crystal, respectively (Table 1)”. (line 142)

General comment for the figures: often the image-quality is too poor and the information in the caption and/or in the text is not sufficient.

The quality of the Figures has been improved, and information in the text are supplemented.

Fig 2A: specify what is the meaning of numbers reported in the legend. Why some numbers are missed? 

Only some curves were shown in figure 2A because showing all 27 curves made it unreadable. We changed it, now there are 2 curves in the figure, one - characteristic for the group of curves from 1 to 12 and second - for the remaining curves

Fig. 2C: very bad quality. Specify in the caption if the curves are, for example, different broken-lines that connect the experimental points.

The quality of the Figures has been improved but also the drawings are changed.

The paragraph 142 – 151 is not clear: it is strongly necessary to improve it.

We change it to:

“The glow curves measured for the Tb-doped samples are shown in Figure 2A. It is visible that these curves differ significantly from each other, and can be divided into two groups regarding to their shape. The first group contains the curves measured for the slices from 1 (beginning of the crystal) to 12, and the second group contains the curves measured for the samples from 13 to the end of the crystal. The characteristic feature of the glow curves measured for the first group is the presence of three, well-separated thermally, peaks with maxima at around 100-125 oC, 180-220 oC, and 325-400 oC. The samples included in the second group exhibit more similar glow curve shape with similar peaks of relatively low amplitudes peaked at around 125 oC and 280 – 325 oC. The different shapes of the measured glow curves may suggest that different mechanisms of impurities incorporation were involved in the growth of different parts of the crystal. These are also correlated with the changes of the energetic distribution of electron traps in consecutive slices of crystal what is strongly connected to the shape of the measured glow curves”. (line 149)

Line 155 “[…] the main and only peak […]”: Fig. 3A shows two peaks, and in any case could be possible also a third component between the two external ones. It is necessary to reconsider the comment and the interpretation

We change it to:

In this case, the most prominent TL peak is located at the temperature range between 300 and 380 oC. (line 165)

The paragraph 158 – 167 is not clear: it is strongly necessary to improve it.

We changed it to:

“Figures 2B and 3B show the TL signal integrated over the temperature range from room temperature to 400 C, calculated for the consecutive slices cut from corresponding crystals (Tb-doped in Fig. 2B and Tm,B-doped in Fig 3B). All samples were weighed before the measurements and each value has been normalized to the sample’s weight. As can be seen in these Figures, the integrated TL signal differs significantly from sample to sample. Interestingly, the most unambiguous results were obtained for the Tb-doped sample. Namely, the initial part of the crystal showed the highest luminescence intensity, up to 4-5 times higher as compared to the signal of samples originating from the end of the crystal (Fig. 2B). Moreover, the distribution of values evaluated for the slices included in the first group (slices from 1 to 12) is very heterogeneous (widely spread) in comparison to the distribution of values evaluated for the samples included in the second group (slices from 13 to the end of the crystal), which is quite uniform. In the case of Tm,B-doped crystal (Fig 3B), significant differences in the TL signal intensities are also visible, however, these cannot be directly correlated to the part (beginning or end) of the crystal – it cannot be surely said that the initial part of the crystal shows higher luminescence intensity than the end of the crystal due to the very large spread of points (increasing trend). Also, there is no such great difference between the beginning and end part of the crystal. This may be related to the differences in Tb and Tm incorporation mechanism in the LMP host matrix”. (line 169)

General comment about the 3.1 section.

Maybe, It could be useful to separate the 3.1 section (TL and OSL) in two sections: a section dedicates to the TL measurements, and the second one for the OSL measurements. However, the interpretation of the results should be improved.

Thank you for this remark. We divided this section into two subsections and we tried to write the text clearer. The figures are also changed to help the reader understand the text.

Figs 5 and 8: indicate, in the captions, what is the meaning of TL(1), .., TL(4),…,TL(8)  in the caption and their link with Figs 6 and 9 respectively.

The captions under the figures have been supplemented with this information

Line 227: “only one high-temperature peak” --> “one evident peak around the temperature of 225 °C”

It was corrected.

Line 227-228: also the peak position changes slightly depending on the samples and the zone of the sample. It is better to add this information in the text.

It was corrected.

In all the manuscript, it is very hard to follow what is the crystal-slice used for the different measure reported in the text or in the figures. It is necessary to find a strategy to improve the reading.

The information about the slice number used for the different measures was given in Table 1.

The style of references seem not in the MDPI format: please check it.

The references have been matched to the Materials standard.

Reviewer 2 Report

The paper describes the production and study of 'crystals' of potential phosphor. 

I would question the relevance of the synthetic method chosen and one small aspect of the experimental detail. In the detail 'doping' is outlined but it is not clear whether for example a proportion of the phosphate is replaced by borate or whether the borate is simply added to the LMP basic mixture. This is important as if it is simply added this will inevitably lead to multiphase products. Which leads to the second important point. While SEM/EDX measurements were clearly shown illustrating the non-uniform distribution of the Ln ions, there was no chemical analyses carried out to show whether the Ln was distributed evenly through the 'crystal'. 

I would also expect to see a 'blank crystal' of LMP with no Ln ions studied to provide the background - or at least the data from a previous publication provided for comparison.

On the method, why is the 'drawn crystal' method appropriate? Would the end product use a powder coated screen or a crystal cut from a larger sample? 

On the subject of 'crystal', this is in itself questionable as the authors do not provide any evidence that the sample is actually a crystal and what it is. There are no diffraction data provided (either X-ray or electron) and hence there may very well be crystalline domains where there is phase segregation. 

The synthesis clearly results in a multiphase product in the SEM, but is this important? If the LMP acts merely as a distribution matrix to hold the distribute Ln material then it does not matter, but if the Ln ions must be a part of the matrix such that the electronic nature of the base material provides a means of excitation then this is clearly a problem that requires addressing.

Ultimately this appears to be a phenomenological study. What happens to the luminescent response when we mix these things together? There is little underpinning evidence to suggest why the observed phenomena or their changes occur. 

Author Response

Dear Editor and Reviewers,

The authors are grateful to the reviewers for all comments. The responses to reviewers' comments are provided below. All changes to the text are marked in red. The article has been linguistically checked.

Reviewer #2

The paper describes the production and study of 'crystals' of potential phosphor.

  1. I would question the relevance of the synthetic method chosen and one small aspect of the experimental detail. In the detail 'doping' is outlined but it is not clear whether for example a proportion of the phosphate is replaced by borate or whether the borate is simply added to the LMP basic mixture. This is important as if it is simply added this will inevitably lead to multiphase products. Which leads to the second important point. While SEM/EDX measurements were clearly shown illustrating the non-uniform distribution of the Ln ions, there was no chemical analyses carried out to show whether the Ln was distributed evenly through the 'crystal'.

Thank you for all your remarks. The idea of studying slices cut from the entire length of the crystal came about because of the observation that different parts of the crystal exhibited different emission of luminescence. It should be emphasized that the LiMgPO4 crystals are very sensitive to ionizing radiation and irradiation with even a small dose causes a strong emission of light, both after thermal and optical excitation. We did not know if doping with Tb or Tm causes the same heterogeneity distribution along the crystal length. Our previous work published in Materials was more specifically about doping and the influence of dopants on the TL signals. The XRD analysis was previously performed and we should refer to this in the text. In the previous experiment [  W. Gieszczyk et al.,. Thermoluminescence enhancement of LiMgPO4 crystal host by Tb3+ and Tm3+ trivalent rare-earths ions co-doping, Materials, 2019, 12, 2861.], the XRD analysis was systematically carried out on a group of 14 LMP crystals: without admixture and with the addition of Tb, Tm and B in different proportions. The research showed that no other phase than LiMgPO4 was observed in both the sample with 1% B and the sample with 10% boron. Trace amounts of phases TbPO4, TmPO4 and MgO were found in the samples LMP:Tb (0.8 mol%) and LMP:Tm (0.8 mol%). The samples were powdered before XRD examination. In this case, XRD analysis of the slices surface should be performed in selected areas and we will try to do it in the future.

We added to the text: „In a previous experiment [26], an X-ray diffraction analysis was systematically conducted in a group of 14 LMP crystals: without admixture and with addition of Tb, Tm, and B in different proportions. The samples were powdered before the analysis. The study showed trace amounts of TbPO4, TmPO4, and MgO phases in Tb-doped LMP (0.8 mol%) and Tm-doped LMP (0.8 mol%), but no phases other than LiMgPO4 were observed in samples doped with B (either 1% or 10%).” (line 95)

  1. I would also expect to see a 'blank crystal' of LMP with no Ln ions studied to provide the background - or at least the data from a previous publication provided for comparison.

You are right, such a comparison should be done and we plan to do so in the future. However, the observations of undoped crystals so far did not indicate large inhomogeneities in the crystal.

  1. On the method, why is the 'drawn crystal' method appropriate? Would the end product use a powder coated screen or a crystal cut from a larger sample?

Of course, there are many methods of material synthesis and crystal growth. We prefer to manufacture the crystals with micro-down pulling because of its usefulness, faster times to obtain crystals, and less input material than in Czochralski method. Our „end” material is the crystal slices which are very convenient in research. Slices cut from the crystal are very sensitive to ionizing radiation and it is their greatest advantage.

  1. On the subject of 'crystal', this is in itself questionable as the authors do not provide any evidence that the sample is actually a crystal and what it is. There are no diffraction data provided (either X-ray or electron) and hence there may very well be crystalline domains where there is phase segregation.

Unfortunately, we do not yet have a handy XRD device for the ongoing checking of the structure of the produced crystals. Previously manufactured crystals were tested several times with XRD and each time we received confirmation that no other phase than LiMgPO4 was detected (in addition to traces of other phases). Regardless of whether the produced material is a crystal or a crystalline domains, it does not matter from the point of view of its use in dosimetric applications. It is so sensitive to radiation that it can be used as a detector. The work was aimed not to obtain perfect crystals, the aim of the work was to study the structure of what we produce to signal problems with heterogeneity to other users of micro-pulling down method.

  1. The synthesis clearly results in a multiphase product in the SEM, but is this important? If the LMP acts merely as a distribution matrix to hold the distribute Ln material then it does not matter, but if the Ln ions must be a part of the matrix such that the electronic nature of the base material provides a means of excitation then this is clearly a problem that requires addressing.

You are definitely right, the form of the material we create does not matter. Even if there is a difference in sensitivity in different parts of the crystal, the material is so sensitive to radiation that it does not matter. Our current research was aimed at systematically understanding the structure along the length of the produced crystals for the first time. If further research on the homogenization of the structure by heat treatment (perhaps by annealing the finished crystals or changing the speed of crystal pulling) then it will be a valuable step.

  1. Ultimately this appears to be a phenomenological study. What happens to the luminescent response when we mix these things together? There is little underpinning evidence to suggest why the observed phenomena or their changes occur.

For a long time, we have studied powders obtained as a result of the synthesis described in the article. The powder samples were homogeneous. The idea of making crystals came about because of melting the material during the crystal growth process reduced the signal fading in time after irradiation. The TL and OSL signals of the powders had a very high fading, compared to them the crystals are characterized by a lower fading.

We also tried to crush the produced crystals and press them to make a pellet or combine them with a polymer to solidify the samples, but all these operations led to a large decrease in its sensitivity.

Dear Editor and Reviewers,

The authors are grateful to the reviewers for all comments. The responses to reviewers' comments are provided below. All changes to the text are marked in red. The article has been linguistically checked.

Reviewer #2

The paper describes the production and study of 'crystals' of potential phosphor.

  1. I would question the relevance of the synthetic method chosen and one small aspect of the experimental detail. In the detail 'doping' is outlined but it is not clear whether for example a proportion of the phosphate is replaced by borate or whether the borate is simply added to the LMP basic mixture. This is important as if it is simply added this will inevitably lead to multiphase products. Which leads to the second important point. While SEM/EDX measurements were clearly shown illustrating the non-uniform distribution of the Ln ions, there was no chemical analyses carried out to show whether the Ln was distributed evenly through the 'crystal'.

Thank you for all your remarks. The idea of studying slices cut from the entire length of the crystal came about because of the observation that different parts of the crystal exhibited different emission of luminescence. It should be emphasized that the LiMgPO4 crystals are very sensitive to ionizing radiation and irradiation with even a small dose causes a strong emission of light, both after thermal and optical excitation. We did not know if doping with Tb or Tm causes the same heterogeneity distribution along the crystal length. Our previous work published in Materials was more specifically about doping and the influence of dopants on the TL signals. The XRD analysis was previously performed and we should refer to this in the text. In the previous experiment [  W. Gieszczyk et al.,. Thermoluminescence enhancement of LiMgPO4 crystal host by Tb3+ and Tm3+ trivalent rare-earths ions co-doping, Materials, 2019, 12, 2861.], the XRD analysis was systematically carried out on a group of 14 LMP crystals: without admixture and with the addition of Tb, Tm and B in different proportions. The research showed that no other phase than LiMgPO4 was observed in both the sample with 1% B and the sample with 10% boron. Trace amounts of phases TbPO4, TmPO4 and MgO were found in the samples LMP:Tb (0.8 mol%) and LMP:Tm (0.8 mol%). The samples were powdered before XRD examination. In this case, XRD analysis of the slices surface should be performed in selected areas and we will try to do it in the future.

We added to the text: „In a previous experiment [26], an X-ray diffraction analysis was systematically conducted in a group of 14 LMP crystals: without admixture and with addition of Tb, Tm, and B in different proportions. The samples were powdered before the analysis. The study showed trace amounts of TbPO4, TmPO4, and MgO phases in Tb-doped LMP (0.8 mol%) and Tm-doped LMP (0.8 mol%), but no phases other than LiMgPO4 were observed in samples doped with B (either 1% or 10%).” (line 95)

  1. I would also expect to see a 'blank crystal' of LMP with no Ln ions studied to provide the background - or at least the data from a previous publication provided for comparison.

You are right, such a comparison should be done and we plan to do so in the future. However, the observations of undoped crystals so far did not indicate large inhomogeneities in the crystal.

  1. On the method, why is the 'drawn crystal' method appropriate? Would the end product use a powder coated screen or a crystal cut from a larger sample?

Of course, there are many methods of material synthesis and crystal growth. We prefer to manufacture the crystals with micro-down pulling because of its usefulness, faster times to obtain crystals, and less input material than in Czochralski method. Our „end” material is the crystal slices which are very convenient in research. Slices cut from the crystal are very sensitive to ionizing radiation and it is their greatest advantage.

  1. On the subject of 'crystal', this is in itself questionable as the authors do not provide any evidence that the sample is actually a crystal and what it is. There are no diffraction data provided (either X-ray or electron) and hence there may very well be crystalline domains where there is phase segregation.

Unfortunately, we do not yet have a handy XRD device for the ongoing checking of the structure of the produced crystals. Previously manufactured crystals were tested several times with XRD and each time we received confirmation that no other phase than LiMgPO4 was detected (in addition to traces of other phases). Regardless of whether the produced material is a crystal or a crystalline domains, it does not matter from the point of view of its use in dosimetric applications. It is so sensitive to radiation that it can be used as a detector. The work was aimed not to obtain perfect crystals, the aim of the work was to study the structure of what we produce to signal problems with heterogeneity to other users of micro-pulling down method.

  1. The synthesis clearly results in a multiphase product in the SEM, but is this important? If the LMP acts merely as a distribution matrix to hold the distribute Ln material then it does not matter, but if the Ln ions must be a part of the matrix such that the electronic nature of the base material provides a means of excitation then this is clearly a problem that requires addressing.

You are definitely right, the form of the material we create does not matter. Even if there is a difference in sensitivity in different parts of the crystal, the material is so sensitive to radiation that it does not matter. Our current research was aimed at systematically understanding the structure along the length of the produced crystals for the first time. If further research on the homogenization of the structure by heat treatment (perhaps by annealing the finished crystals or changing the speed of crystal pulling) then it will be a valuable step.

  1. Ultimately this appears to be a phenomenological study. What happens to the luminescent response when we mix these things together? There is little underpinning evidence to suggest why the observed phenomena or their changes occur.

For a long time, we have studied powders obtained as a result of the synthesis described in the article. The powder samples were homogeneous. The idea of making crystals came about because of melting the material during the crystal growth process reduced the signal fading in time after irradiation. The TL and OSL signals of the powders had a very high fading, compared to them the crystals are characterized by a lower fading.

We also tried to crush the produced crystals and press them to make a pellet or combine them with a polymer to solidify the samples, but all these operations led to a large decrease in its sensitivity.

Dear Editor and Reviewers,

The authors are grateful to the reviewers for all comments. The responses to reviewers' comments are provided below. All changes to the text are marked in red. The article has been linguistically checked.

Reviewer #2

The paper describes the production and study of 'crystals' of potential phosphor.

  1. I would question the relevance of the synthetic method chosen and one small aspect of the experimental detail. In the detail 'doping' is outlined but it is not clear whether for example a proportion of the phosphate is replaced by borate or whether the borate is simply added to the LMP basic mixture. This is important as if it is simply added this will inevitably lead to multiphase products. Which leads to the second important point. While SEM/EDX measurements were clearly shown illustrating the non-uniform distribution of the Ln ions, there was no chemical analyses carried out to show whether the Ln was distributed evenly through the 'crystal'.

Thank you for all your remarks. The idea of studying slices cut from the entire length of the crystal came about because of the observation that different parts of the crystal exhibited different emission of luminescence. It should be emphasized that the LiMgPO4 crystals are very sensitive to ionizing radiation and irradiation with even a small dose causes a strong emission of light, both after thermal and optical excitation. We did not know if doping with Tb or Tm causes the same heterogeneity distribution along the crystal length. Our previous work published in Materials was more specifically about doping and the influence of dopants on the TL signals. The XRD analysis was previously performed and we should refer to this in the text. In the previous experiment [  W. Gieszczyk et al.,. Thermoluminescence enhancement of LiMgPO4 crystal host by Tb3+ and Tm3+ trivalent rare-earths ions co-doping, Materials, 2019, 12, 2861.], the XRD analysis was systematically carried out on a group of 14 LMP crystals: without admixture and with the addition of Tb, Tm and B in different proportions. The research showed that no other phase than LiMgPO4 was observed in both the sample with 1% B and the sample with 10% boron. Trace amounts of phases TbPO4, TmPO4 and MgO were found in the samples LMP:Tb (0.8 mol%) and LMP:Tm (0.8 mol%). The samples were powdered before XRD examination. In this case, XRD analysis of the slices surface should be performed in selected areas and we will try to do it in the future.

We added to the text: „In a previous experiment [26], an X-ray diffraction analysis was systematically conducted in a group of 14 LMP crystals: without admixture and with addition of Tb, Tm, and B in different proportions. The samples were powdered before the analysis. The study showed trace amounts of TbPO4, TmPO4, and MgO phases in Tb-doped LMP (0.8 mol%) and Tm-doped LMP (0.8 mol%), but no phases other than LiMgPO4 were observed in samples doped with B (either 1% or 10%).” (line 95)

  1. I would also expect to see a 'blank crystal' of LMP with no Ln ions studied to provide the background - or at least the data from a previous publication provided for comparison.

You are right, such a comparison should be done and we plan to do so in the future. However, the observations of undoped crystals so far did not indicate large inhomogeneities in the crystal.

  1. On the method, why is the 'drawn crystal' method appropriate? Would the end product use a powder coated screen or a crystal cut from a larger sample?

Of course, there are many methods of material synthesis and crystal growth. We prefer to manufacture the crystals with micro-down pulling because of its usefulness, faster times to obtain crystals, and less input material than in Czochralski method. Our „end” material is the crystal slices which are very convenient in research. Slices cut from the crystal are very sensitive to ionizing radiation and it is their greatest advantage.

  1. On the subject of 'crystal', this is in itself questionable as the authors do not provide any evidence that the sample is actually a crystal and what it is. There are no diffraction data provided (either X-ray or electron) and hence there may very well be crystalline domains where there is phase segregation.

Unfortunately, we do not yet have a handy XRD device for the ongoing checking of the structure of the produced crystals. Previously manufactured crystals were tested several times with XRD and each time we received confirmation that no other phase than LiMgPO4 was detected (in addition to traces of other phases). Regardless of whether the produced material is a crystal or a crystalline domains, it does not matter from the point of view of its use in dosimetric applications. It is so sensitive to radiation that it can be used as a detector. The work was aimed not to obtain perfect crystals, the aim of the work was to study the structure of what we produce to signal problems with heterogeneity to other users of micro-pulling down method.

  1. The synthesis clearly results in a multiphase product in the SEM, but is this important? If the LMP acts merely as a distribution matrix to hold the distribute Ln material then it does not matter, but if the Ln ions must be a part of the matrix such that the electronic nature of the base material provides a means of excitation then this is clearly a problem that requires addressing.

You are definitely right, the form of the material we create does not matter. Even if there is a difference in sensitivity in different parts of the crystal, the material is so sensitive to radiation that it does not matter. Our current research was aimed at systematically understanding the structure along the length of the produced crystals for the first time. If further research on the homogenization of the structure by heat treatment (perhaps by annealing the finished crystals or changing the speed of crystal pulling) then it will be a valuable step.

  1. Ultimately this appears to be a phenomenological study. What happens to the luminescent response when we mix these things together? There is little underpinning evidence to suggest why the observed phenomena or their changes occur.

For a long time, we have studied powders obtained as a result of the synthesis described in the article. The powder samples were homogeneous. The idea of making crystals came about because of melting the material during the crystal growth process reduced the signal fading in time after irradiation. The TL and OSL signals of the powders had a very high fading, compared to them the crystals are characterized by a lower fading.

We also tried to crush the produced crystals and press them to make a pellet or combine them with a polymer to solidify the samples, but all these operations led to a large decrease in its sensitivity.

Dear Editor and Reviewers,

The authors are grateful to the reviewers for all comments. The responses to reviewers' comments are provided below. All changes to the text are marked in red. The article has been linguistically checked.

Reviewer #2

The paper describes the production and study of 'crystals' of potential phosphor.

  1. I would question the relevance of the synthetic method chosen and one small aspect of the experimental detail. In the detail 'doping' is outlined but it is not clear whether for example a proportion of the phosphate is replaced by borate or whether the borate is simply added to the LMP basic mixture. This is important as if it is simply added this will inevitably lead to multiphase products. Which leads to the second important point. While SEM/EDX measurements were clearly shown illustrating the non-uniform distribution of the Ln ions, there was no chemical analyses carried out to show whether the Ln was distributed evenly through the 'crystal'.

Thank you for all your remarks. The idea of studying slices cut from the entire length of the crystal came about because of the observation that different parts of the crystal exhibited different emission of luminescence. It should be emphasized that the LiMgPO4 crystals are very sensitive to ionizing radiation and irradiation with even a small dose causes a strong emission of light, both after thermal and optical excitation. We did not know if doping with Tb or Tm causes the same heterogeneity distribution along the crystal length. Our previous work published in Materials was more specifically about doping and the influence of dopants on the TL signals. The XRD analysis was previously performed and we should refer to this in the text. In the previous experiment [  W. Gieszczyk et al.,. Thermoluminescence enhancement of LiMgPO4 crystal host by Tb3+ and Tm3+ trivalent rare-earths ions co-doping, Materials, 2019, 12, 2861.], the XRD analysis was systematically carried out on a group of 14 LMP crystals: without admixture and with the addition of Tb, Tm and B in different proportions. The research showed that no other phase than LiMgPO4 was observed in both the sample with 1% B and the sample with 10% boron. Trace amounts of phases TbPO4, TmPO4 and MgO were found in the samples LMP:Tb (0.8 mol%) and LMP:Tm (0.8 mol%). The samples were powdered before XRD examination. In this case, XRD analysis of the slices surface should be performed in selected areas and we will try to do it in the future.

We added to the text: „In a previous experiment [26], an X-ray diffraction analysis was systematically conducted in a group of 14 LMP crystals: without admixture and with addition of Tb, Tm, and B in different proportions. The samples were powdered before the analysis. The study showed trace amounts of TbPO4, TmPO4, and MgO phases in Tb-doped LMP (0.8 mol%) and Tm-doped LMP (0.8 mol%), but no phases other than LiMgPO4 were observed in samples doped with B (either 1% or 10%).” (line 95)

  1. I would also expect to see a 'blank crystal' of LMP with no Ln ions studied to provide the background - or at least the data from a previous publication provided for comparison.

You are right, such a comparison should be done and we plan to do so in the future. However, the observations of undoped crystals so far did not indicate large inhomogeneities in the crystal.

  1. On the method, why is the 'drawn crystal' method appropriate? Would the end product use a powder coated screen or a crystal cut from a larger sample?

Of course, there are many methods of material synthesis and crystal growth. We prefer to manufacture the crystals with micro-down pulling because of its usefulness, faster times to obtain crystals, and less input material than in Czochralski method. Our „end” material is the crystal slices which are very convenient in research. Slices cut from the crystal are very sensitive to ionizing radiation and it is their greatest advantage.

  1. On the subject of 'crystal', this is in itself questionable as the authors do not provide any evidence that the sample is actually a crystal and what it is. There are no diffraction data provided (either X-ray or electron) and hence there may very well be crystalline domains where there is phase segregation.

Unfortunately, we do not yet have a handy XRD device for the ongoing checking of the structure of the produced crystals. Previously manufactured crystals were tested several times with XRD and each time we received confirmation that no other phase than LiMgPO4 was detected (in addition to traces of other phases). Regardless of whether the produced material is a crystal or a crystalline domains, it does not matter from the point of view of its use in dosimetric applications. It is so sensitive to radiation that it can be used as a detector. The work was aimed not to obtain perfect crystals, the aim of the work was to study the structure of what we produce to signal problems with heterogeneity to other users of micro-pulling down method.

  1. The synthesis clearly results in a multiphase product in the SEM, but is this important? If the LMP acts merely as a distribution matrix to hold the distribute Ln material then it does not matter, but if the Ln ions must be a part of the matrix such that the electronic nature of the base material provides a means of excitation then this is clearly a problem that requires addressing.

You are definitely right, the form of the material we create does not matter. Even if there is a difference in sensitivity in different parts of the crystal, the material is so sensitive to radiation that it does not matter. Our current research was aimed at systematically understanding the structure along the length of the produced crystals for the first time. If further research on the homogenization of the structure by heat treatment (perhaps by annealing the finished crystals or changing the speed of crystal pulling) then it will be a valuable step.

  1. Ultimately this appears to be a phenomenological study. What happens to the luminescent response when we mix these things together? There is little underpinning evidence to suggest why the observed phenomena or their changes occur.

For a long time, we have studied powders obtained as a result of the synthesis described in the article. The powder samples were homogeneous. The idea of making crystals came about because of melting the material during the crystal growth process reduced the signal fading in time after irradiation. The TL and OSL signals of the powders had a very high fading, compared to them the crystals are characterized by a lower fading.

We also tried to crush the produced crystals and press them to make a pellet or combine them with a polymer to solidify the samples, but all these operations led to a large decrease in its sensitivity.

Round 2

Reviewer 1 Report

The current version of the work is comprehensive, if not exhaustive, well written, and useful for specialists in this field. Publication is recommended, without need for significant revisions. But it is strongly encouraged to improve the quality of figures 2,3,4,5,6.

Line 40:  “The method provides also”  à “The method also provides”

Line 49: at the end of “Research is currently ongoing to elucidate this effect” could be interest add some references.

Line 144: (Table 1).-> (see Table 1).

Figures 2, 3, 4, 5, 6: the values ​​on the y-axis are not legible: it is necessary to improve the quality of the images.

Author Response

Dear Reviewer,

Thank you for your comments. Corrections were made to the text of the manuscript. The Figures have been corrected. Regarding the following note:

„at the end of “Research is currently ongoing to elucidate this effect” could be interest add some references”.

Unfortunately we do not have any published work to cite here yet.

Your sincerely,

Barbara Marczewska